# Scaling-up molecular logic to meso-systems via self-assembly

Ze-Qing Chen [1], Brian Daly[1], Chao-Yi Yao[2], Hannah S. N. Crory[1], Yikai Xu [1,3], Ziwei Ye [1], H. Q. Nimal Gunaratne[1], Ayumi Kimura [4], Seiichi Uchiyama [5], Steven E. J. Bell [1], Eric V. Anslyn [1,6] & A. Prasanna de Silva [1] ✉

Due to the small size and biocompatibility of molecules, molecular logic-based computation is a gateway to the informational basis of life processes. Logic-based computation operates widely with discrete molecules of up to nanometric sizes. The contribution of molecule-based bulk materials of milli/centimetric size to the field has begun in more recent years. However, artificial molecule-based meso-scale systems which intrinsically perform logic operations are very rare. Here, we show that self-assembled systems consisting of cyclophane octacarboxylates and a cationic surfactant can perform such functions, where a membrane itself behaves as a Reset-Set Flip-Flop which is integrated with 7 more logic elements. Now that molecular logic-based computation operates across a wide range of contiguous size-scales, the way opens for its general use in information processing aspects of biology and synthetic biology.

The informational basis of life is illustrated by computational operations performed by molecules with various levels of organization, e.g. a biomolecule itself (nanometric), a cell (micrometric) or a brain (decimetric)[1–7]. Molecular logic[8–21] sheds light on these bioprocesses. For instance, edge detection in the visual attention process is performed by groups of glial cells in the retina[22]. This has been emulated at the level of bacteria[23], biomolecular systems[24] and small molecules[25,26] via logic schemes. We now show that a membrane system is switchable in a Boolean manner involving more than 10 logic elements. This would be important as a way of scaling logic operations from the molecular- to the meso-dimension. For the present purpose, the most convenient definition of meso-scale is the scale of sizes larger than nanometric and smaller than micrometric. Although a few molecular-scale logic devices have been embedded in membranes[27–29] and some devices have been based on molecular polymeric materials[30–34], membrane logic is very rare[35]. Conditional building-up and breakdown of membranes and proteins occurs continuously in nature, as seen during autophagy[36] for

instance. However, we are not aware of any Boolean schemes being identified in these situations. We also show that logical complexity can be increased when molecules assemble and organize into a membrane. We do this by partnering anionic *p*-cyclophanes like **1-5**[37–43] with a cationic detergent **10**[44] for binding/unbinding studies. Anionic *p*-cyclophanes and relatives[37–43,45–50] have a track record of binding organic cations in water[51], e.g. relatives of **1**[37] capture steroidal cations[49]. A few of these[39–43,50] can be shape-switched by electrochemical and chemical stimuli to unbind the guests in a logical manner[1–21].

## Results

### Synthesis

We approach *p*-cyclophane octacarboxylate **1** (Fig. 1A) via a convenient route (Supplementary methods 1, Supplementary Figs. 1, 2a-2f & Supplementary note 1) which is different from Koga's original synthesis involving a close relative of ethyl ester **2**[37]. Cyclophane

[1]School of Chemistry and Chemical Engineering, Queen's University, Belfast BT9 5AG Northern Ireland, UK. [2]School of Chemistry and Chemical Engineering, Central South University, Yuelu District, Changsha, Hunan Province 410006, China. [3]Key Laboratory for Advanced Materials and Feringa Nobel Prize Scientist Joint Research Center, Frontiers Science Center for Materiobiology and Dynamic Chemistry, School of Chemistry and Molecular Engineering, East China University of Science and Technology, Shanghai 200237, China. [4]Institute of Engineering Innovation, The University of Tokyo, 2-11-16 Yayoi, Bunkyo-ku, Tokyo 113-8656, Japan. [5]Graduate School of Pharmaceutical Sciences, The University of Tokyo, 7-3-1 Hongo, Bunkyo-ku, Tokyo 113-0033, Japan. [6]Department of Chemistry, University of Texas at Austin, 100 E 24th Street, Norman Hackerman Building, Austin, TX 78712, USA. ✉e-mail: a.desilva@qub.ac.uk

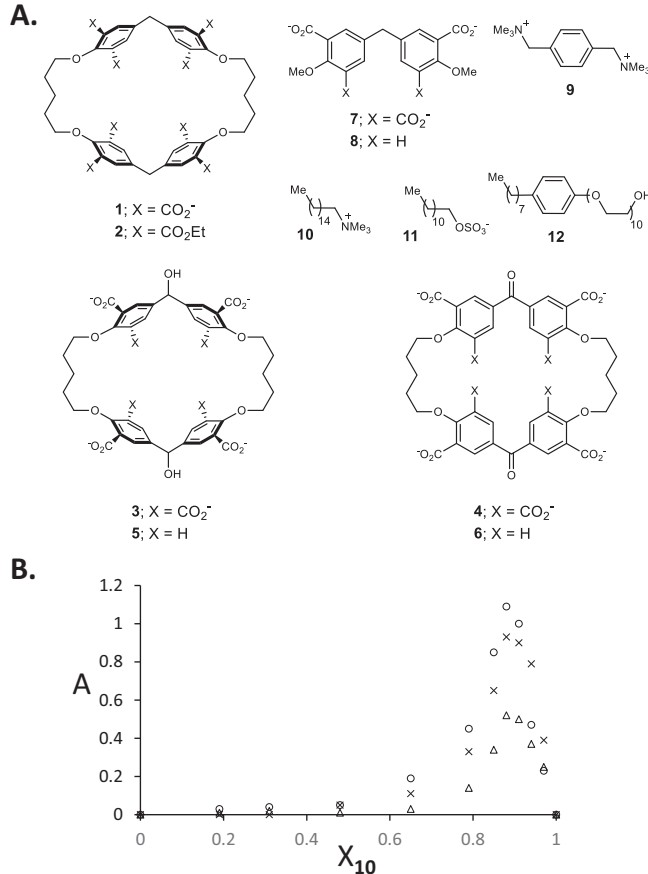

**A.**

7; X = CO2⁻
8; X = H

9

1; X = CO2⁻
2; X = CO2Et

10    11    12

3; X = CO2⁻
5; X = H

4; X = CO2⁻
6; X = H

**B.**

**Fig. 1 | Chemical compounds and turbidity-mole fraction plots. A** Structures of the molecules employed in this work. The counterions for **1, 3-8** and **11** are Na⁺, for **9** is Br⁻ and for **10** is Cl⁻. **B** Turbidity (as measured by absorbance (A) at 500 nm with 1 cm path length) versus mole fraction of **10** ($X_{10}$) in mixture with **1** (circles), **4** (triangles) and **3** (crosses) in water.

diketoneoctacarboxylate **4** is obtained by KMnO₄ oxidation of **1**. NaBH₄ reduction of **4** produces cyclophane dialcoholoctacarboxylate **3**. **3** can be oxidized back to **4**. Cyclophane tetracarboxylates **5** and **6**[39], as well as the control compounds **7** and **8**[52–56], are prepared according to literature procedures (Supplementary methods 2 & Supplementary Fig. 3). Guest **9** and relatives have been used previously for studies in supramolecular chemistry[38,39]. Detergents **10**-**12** are commercially available.

### Physical measurements

An interaction between cyclophane octacarboxylate **1** and cationic detergent **10** is immediately signaled by a turbidity in basic aqueous solution when they are mixed (Supplementary note 2 & Supplementary Fig. 4). For the present purpose, the most convenient definition of turbidity is in terms of absorption of light (at a sufficiently long wavelength) caused by the scattering of light by the suspended particles. Since our compounds do not absorb light in the visible region, a wavelength of 500 nm was found to be practical. This turbidity effect maximizes at ca. 1:8 (**1**:**10**) stoichiometry (Fig. 1B). Similar results are found when cyclophane octacarboxylates **3** and **4** are combined with cationic detergent **10**. Replacement of **10** by anionic **11** or nonionic **12** detergents does not result in turbid solutions. Mixtures of **10** and cyclophanes with fewer carboxylate units (**5** and **6**) or acyclic control compounds (**7** and **8**) also show no turbidity. So, the formation of a meso-scale membrane capable of scattering visible light appears to require a macrocycle with sufficient carboxylate units on both faces. Dynamic light scattering data allows an estimate of the size of these

assemblies. The particle size distribution maximizes at 179, 194 and 615 nm for cyclophanes **1**, **4** and **3** respectively, when mixed with detergent **10** (Supplementary note 3 & Supplementary Fig. 5a–c). However, the distribution is skewed somewhat towards larger particles in the case of cyclophane **3**, perhaps because it is noticeably less soluble in basic water than cyclophanes **1** and **4**. Broadly similar results are obtained from nanoparticle tracking analysis (Supplementary note 4 & Supplementary Fig. 6a–c).

The lack of interactions between cyclophanes **1**, **4** and **3** and detergents **11** and **12** is confirmed by the critical micelle concentrations of **11** and **12** in water being essentially undisturbed by the cyclophanes (Supplementary note 5 & Supplementary Table 1). In contrast, the critical aggregation concentration of detergent **10** drops by an order of magnitude in the presence of cyclophanes **1**, **4** and **3** (Supplementary note 5 & Supplementary Table 1).

Importantly, transmission electron microscopy of the turbid aqueous solutions with MoO₄²⁻ staining shows fragments of multi-layered lamellae (Fig. 2A, Supplementary note 6 & Supplementary Fig. 7a–c). These can be understood to have originated from the self-assembly process schematized in Fig. 2C. Cyclophane octacarboxylate **1** is known[37–39] to orient its phenylene rings orthogonal to the mean macrocycle plane. Ion-pairing of octacarboxylate **1** with 8 molecules of **10** produces the $1 \cdot 10_8$ unit, where the alkyl chains of **10** are approximately orthogonal to the mean macrocycle plane. These units can line up axially (relative to the mean macrocycle plane) and polymerize supramolecularly using hydrophobic interactions and some degree of interdigitation of alkyl chains[57,58], as commonly found in cell membranes[59]. Having four alkyl chains on each face of the cyclophane offers a high probability for such alkyl chain interdigitation and close-packing even if one of those chains is included inside the cyclophane cavity. The columns arising in this way stack laterally to give multi-layered lamellae, akin to a bamboo thicket (Fig. 2B). Staining of these lamellae, by MoO₄²⁻ units which associate at cationic sites, leads to the fragments which are observed (Fig. 2A). Cyclophane dialcoholocta-carboxylate **3** has a very similar conformation to **1** because of the sp³ hybridized carbon between the phenylene groups, and has very similar ability to achieve the self-assembly process of Fig. 2C. Cyclophane diketoneoctacarboxylate **4** has a conformation where the phenylene groups flatten into the mean macrocycle plane due to the sp² hybridized carbon between these groups[39]. Still, the availability of four alkyl chains on each face of the cyclophane allows axial stacking of $4 \cdot 10_8$ units. The different abilities of **3** and **4** to produce multi-layered assemblies with detergent **10** under certain conditions due to the different arrangements of the alkyl chains (Fig. 3A) will be demonstrated in a later section. Ion-pairing of detergent **10** with either cyclophane tetracarboxylate **5** or **6** cannot provide an axially stackable unit with similar ability for alkyl chain interdigitation. Control compounds **7** and **8** are even worse in this regard, since they lack the pre-organization ability of a macrocycle.

Quantification of the interlayer spacings observed in the multi-layered lamellar fragments shown in Fig. 2A gives $3.2 \pm 0.3$, $3.0 \pm 0.1$ and $2.4 \pm 0.4$ nm for the combination of cationic detergent **10** with octacarboxylate **1**, dialcoholoctacarboxylate **3** and diketoneocta-carboxylate **4** respectively. Clearly, the membranes arising from the two cyclophanes **1**&**3** with phenylene rings oriented orthogonal to the mean macrocycle plane have essentially identical interlayer spacings of 3.1 nm. In contrast, the membrane generated from the cyclophane **4** with phenylene rings flattened into the mean macrocycle plane has a significantly smaller interlayer spacing of 2.4 nm.

*p*-Xylyldiammonium dications, e.g. **9**, are known to nest inside the cavity of cyclophane dialcoholtetracarboxylates like **5**[39], because of their geometric complementarity and because of favorable hydrophobic and electrostatic interactions. In contrast, the cavity of cyclophane diketonetetracarboxylates like **6**[39], is too small, owing to the phenylene groups flattening into the mean macrocycle plane (*vide*

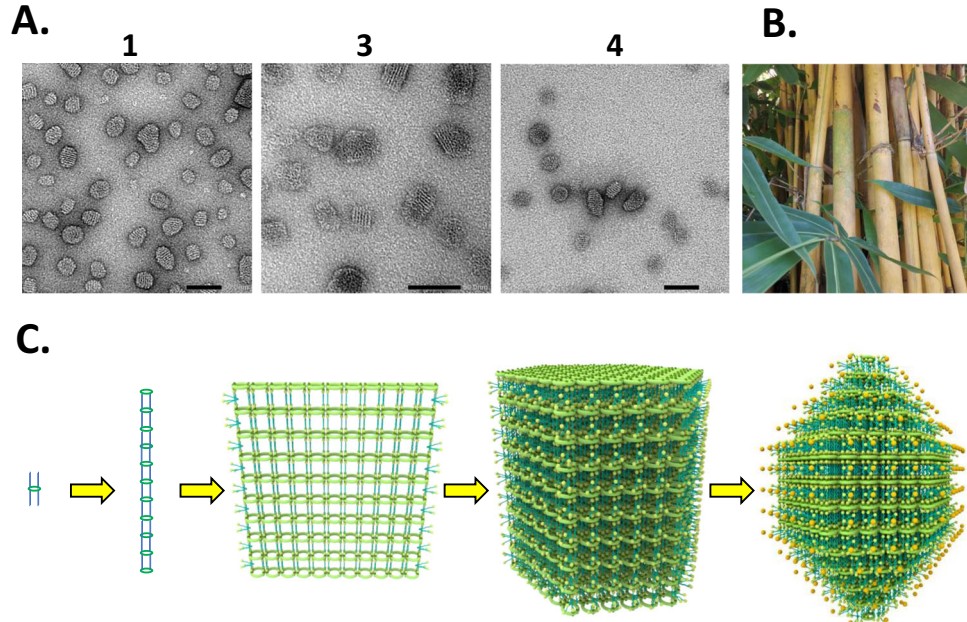

**Fig. 2 | Transmission electron microscopy and the bamboo thicket model of self-assembly. A** Transmission electron micrographs for assemblies of **10** formed with **1, 3** and **4** in water, stained with $MoO_4^{2-}$. Positively charged stains such as $Gd^{3+}$ are unsuccessful. The scale bars are 50 nm. **B** A bamboo thicket. **C** Schematic representation of the self-assembly of **10** with any of the cyclophanes **1, 3** or **4** via supramolecular polymerization of the **1·10₈** unit, for example. The resulting multilayered lamellae, reminiscent of a bamboo thicket, are degraded into multi-layered fragments during staining for electron microscopy. $MoO_4^{2-}$ is represented by orange spheres.

*supra*), so that binding of **9** is not seen. A parallel behavior is found when **9** is mixed with cyclophanes **1, 3** & **4**. The evidence is found in the corresponding $^1H$ NMR spectra (Fig. 3B, Supplementary note 7 & Supplementary Fig. 8a–c), where the complexation-induced chemical shift differences (Δδ) are much smaller for the pair of **9** and **4**, as compared to the other guest-host pairs. Binding constants (log β) of 3.6 are found from concentration-dependent studies for complexes **9·1** and **9·3** in water, whereas the corresponding value for the putative complex **9·4** is <2.

Now we consider the disruption of the membranes arising from each member of the redox-pair of cyclophanes **4** (diketone) and **3** (dialcohol). Turbid aqueous solutions of **3·10₈** are destabilized by the gradual addition of the competing guest xylyldiammonium **9**, which results in clarification characterized by a mid-point of $10^{-4.1}$ M of **9** (Fig. 4A). This mid-point concentration value of **9** for the depolymerization is significantly smaller than what would be indicated ($10^{-3.6}$ M) by the binding constant (logβ=3.6) for **9**'s interaction with the cyclophane **3**. Essentially the same mid-point concentration of **9** is found when the model cyclophane **1** is examined in the same way, as might be anticipated from the similar orientation of phenylene units in both cases. When an analogous experiment is conducted with the diketone macrocycle with collapsed phenylene 'walls', i.e. with turbid solutions of **4·10₈**, clarification occurs at ca. $10^{-5}$ M of **9**, even though **9** interacts with cyclophane **4** hardly at all even at concentrations of $10^{-3}$ M. This suggests that the destabilization of these membranes by **9** occurs without initial association with the cyclophane units, which would be rather inaccessible within the membrane anyway. This is also not a salt effect, since the addition of NaCl causes essentially the same salt-induced turbidity changes in membranes arising from each of the three cyclophanes **1, 3** and **4**. These changes are also bimodal and occur at much higher salt concentrations of ca. $>10^{-1.5}$ M (Supplementary note 8 & Supplementary Fig. 9). The relative hydrophobicity of dicationic **9** would allow it to break up the interdigitation of cyclophane-bound **10** by binding at this rather hydrophobic location, while providing some cationic amphiphilic action of its own (Fig. 3C),

so that the size of the membrane particles decrease below that which would scatter visible light. Several membranes based on other macrocycles such as pillararenes which are switchable with gases[60–62], ions[61,63] or small molecules[61] via different mechanisms are known. However, these membranes are mostly mono- or bi-layered vesicles, and are not the rare multi-layered lamellae (Fig. 2A) observed in the present work. Following photoinduced electron transfer to chlorinated solvents, certain triphenylamines outfitted with amide units and long alkyl chains give rise to fibrils via supramolecular polymerization[64]. Since these are thermally reversible, they are also relevant examples of switchable meso-scale systems. It is also relevant to note a report[65] where the concept of Boolean drug release following depolymerization of materials[33,34] has been miniaturized to the nanoparticle level. However, the nanoparticles are irreversibly degraded during the release process.

## Discussion

Building a memory with two states begins with a ketone **4** and an alcohol **3** which are interconvertible by chemical redox reactions. Since the redox pair **4/3** is structurally elaborated into two cyclophane macrocycles, the function of switchable binding/unbinding of guests comes into view. Employment of surfactant **10** as the guest introduces membrane formation into the aqueous system. The transmission electron microscopy results which gave an interlayer spacing of 2.4 nm for the system containing **4**&**10** and a spacing of 3.1 nm for the case involving **3**&**10** can be analyzed further.

The 'height' of the phenylene units of cyclophanes **3** or **1** (perpendicular to the mean macrocycle plane) can be approximated by the appropriate oxygen-oxygen interatomic distance in isophthalate[66] (0.74 nm) plus the van der Waals radii of the termini (0.15 nm each)[67]. Similarly, the 'height' of the flattened phenylene units of cyclophane diketone **4** can be approximated by the thickness of an aromatic pi-electron cloud (0.34 nm)[68], although this might be tempered by transannular steric interactions. The difference between these two sets of phenylene unit 'heights' is 0.7 nm. Remarkably, this is identical to

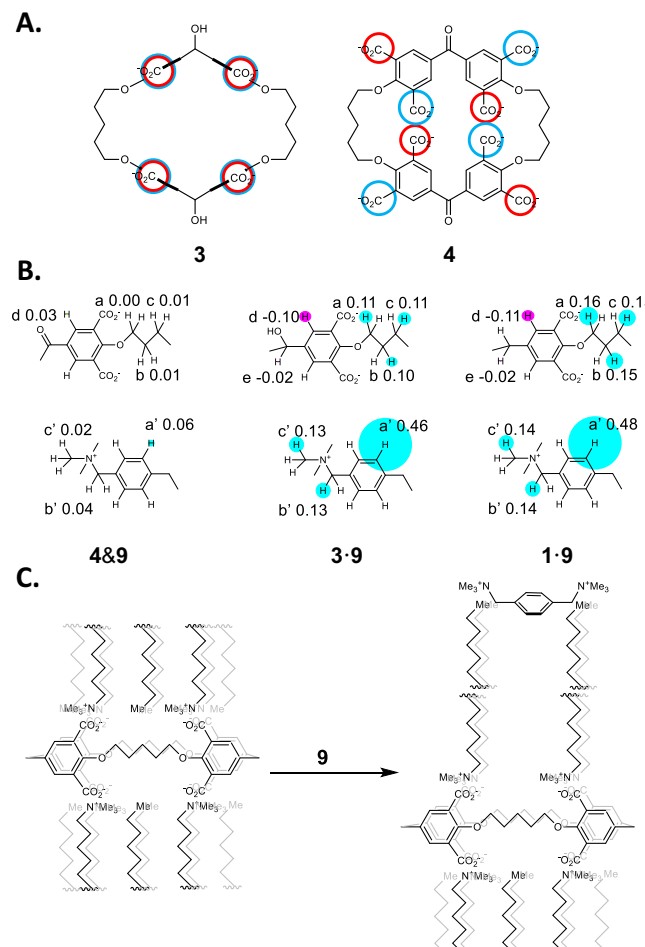

**Fig. 3 | Hydrocarbon chain arrangements within the self-assembly, complexation-induced chemical shift differences and mode of membrane disruption. A** View orthogonal to the mean macrocycle plane of the arrangement of alkyl chains of **10** bound to cyclophanes **3** and **4**. Chains pointing up or down are symbolized by blue or red circles respectively. **B** $\Delta\delta$ maps arising from $^1$H NMR spectra of **4&9**, **3·9** and **1·9** respectively. -$\Delta\delta$ values are noted on the partial molecular structures. Relative magnitudes of $\Delta\delta$ values are shown by the radii of circles centered on one of the appropriate protons. Signs of $\Delta\delta$ values, whether negative or positive, are symbolized by blue or red circles respectively. **C 9**-induced depolymerization of poly(**1·10$_8$**).

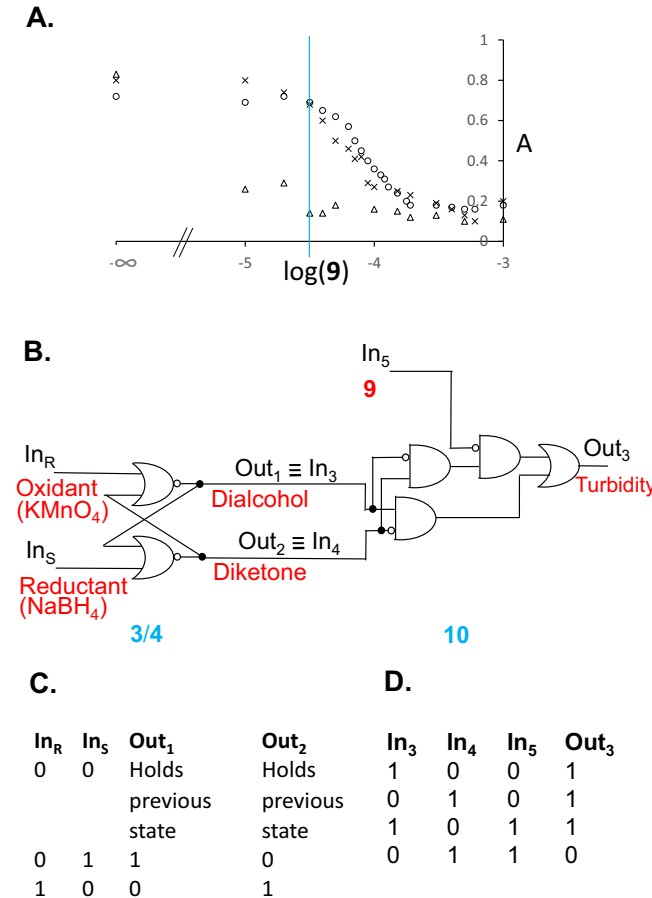

**Fig. 4 | Dose-dependent membrane disruption plots, corresponding logic gate array and truth tables. A** Turbidity (as measured by absorbance (A) at 500 nm with 1 cm path length) versus log(**9**) for **10** in mixture with **1** (circles), **4** (triangles) and **3** (crosses) in water. For ease of comparison, the turbidity values for the case involving **3** are plotted after dividing by 2. **B** Physical electronic representation of a RS flip-flop integrated with a substantial array of logic elements corresponding to the **9**-induced reversible membrane formation/collapse by redox pair **3/4** in the presence of **10**. **C, D** Truth tables for the sequential and combinational logic components involved in part B. Input$_5$ (**9**) is taken as 'high' (1) when its concentration is $10^{-4.5}$ M, and 'low' (0) when it is absent. Output$_3$, the turbidity when (**9**) = $10^{-4.5}$ M, is taken as 'high' (1) when the absorbance is >0.4. Otherwise, Output$_3$ is 'low' (0).

the difference between the observed interlayer spacing of the membranes arising from cyclophanes **3** (or **1**) and cyclophane diketone **4**. This fits the model of Fig. 2C if the degree of alkyl chain interdigitation of **10** and the orientation of those alkyl chains with respect to the mean macrocycle plane are constant across all three cases.

**10**, in its extended form, occupies a length of 2.50 nm[69]. So, the interlayer spacing of the membranes arising from cyclophanes **1&3** as partners can be calculated to be 2.50 + 1.04 = 3.54 nm for full interdigitation of alkyl chains. The observed value of 3.1 nm can be ascribed to some coiling of the chains of **10**. A similar consideration applies to the case concerning cyclophane diketone **4**.

The significantly higher stability of the membrane based on cyclophane dialcohol **3**, c.f. that based on diketone **4**, in the face of disruption by **9**, can be attributed to the erect phenylene walls producing nanotubes of regular diameter with the eight hydrocarbon chains of surfactant **10** being positioned equidistant from the macrocycle axis leading to strong interdigitation and close-packing (Fig. 3A). Notably, all eight carboxylates of **3** are equidistant from the macrocycle axis, and these pair with the cationic trimethylammonium

headgroups of the eight copies of **10**. In contrast, the collapsed phenylene walls of cyclophane diketone **4** would split the alkyl chains into two groups proximal and distal to the macrocycle axis (Fig. 3A), which would be less conducive to efficient alkyl chain interdigitation and close-packing. In this instance, there would be four carboxylates which are close to the macrocycle axis and four outside the macrocycle frame. Steric considerations orient the neighboring carboxylates of the former set towards opposite faces of the macrocycle, which control the orientations of the latter set in turn.

Previously, we have shown that redox interconversions of the simple alcohol-ketone system emulate the features of a computer memory element, the RS Flip-Flop[39–43,70]. Once reduced (reductant Input$_S$ = 1), the alcohol is maintained (Output$_1$ alcohol state = 1) whether no reductant (reductant Input$_S$ = 0) or excess reductant (reductant Input$_S$ = 1) is applied subsequently. Once oxidized (oxidant Input$_R$ = 1), the ketone remains the only product (Output$_2$ ketone state = 1) whether no oxidant (oxidant Input$_R$ = 0) or excess oxidant (oxidant Input$_R$ = 1) is applied thereafter. By building dialcohol-diketone macrocyclic host systems, we could integrate various downstream logic

gates with the RS Flip-Flop, depending on the property being exhibited by the molecular device. The **9**-induced switching behavior of the turbidity output in the current work leads to the most complex logic gate array seen thus far. When Input$_5$ (**9**) is considered to be 'high' (1) when its concentration is $10^{-4.5}$ M and to be 'low' (0) when it is absent, the behavior described above can be accommodated in the truth tables in Fig. 4C, D. Analysis of this table according to the 'sum of products' approach of Boolean logic analysis[71], gives rise to the minimized gate array shown in Fig. 4B. It contains 7 logic elements downstream from the RS Flip-Flop. Although the redox-interconvertible **4**/**3** system only displays RS Flip-Flop logic behavior, its self-assembly with detergent **10** produces a more complex logic action when switched by additive **9**. When **9** is absent, and since only dialcohol or diketone can exist at a given moment, i.e. they can't both be present and they can't both be absent, the downstream gate array simplifies first to a XOR and finally to an OR gate (Supplementary note 9).

A potential use of our findings on membrane logic systems would be in the diagnosis of membrane-related diseases[72] just as nanometric logic systems were proposed for diseases concerning electrolytes[73,74] and commercialized later for blood diagnostics (https://www.optimedical.com). However, the significance of our findings for biological systems needs to be tempered because these experiments were conducted in basic aqueous solution. Furthermore, chemically-switchable logic systems accumulate chemical waste. However, there are examples where waste does not interfere with their operation at least in the short term[40]. Ref. 40 is also an example closely related to the present work where three cycles of operation have been run successfully. However, the present work is currently validated for single use. A further limitation is introduced by the fact that the redox interconversions of **3** and **4** are not quantitative at present. The sharpness of switching of turbidity (Fig. 4A) will be tempered somewhat owing to this.

In conclusion, the molecular-level redox interconversion between a cyclophane-based dialcohol and a diketone has been transformed into the differential binding of a cationic surfactant within multilayered lamellar superstructures of submicrometer size. The latter assemblies are maintained or collapsed in the presence of a small aromatic dication under specified aqueous conditions. Such switching 'on' or 'off' at the meso-scale corresponds to the integration of a molecular memory with a substantial array of combinational logic gates. In contrast, the guest binding/unbinding output of the cyclophane dialcohol/diketone system on its own behaves only as a RS Flip-Flop at the molecular level. Here, the scaling-up in size also increases the complexity of the logic gate array which is displayed.

## Methods

### Synthesis and characterization of cyclophanes leading to membranes

All starting materials were purchased from Sigma-Aldrich. All procedures are given in the supplementary methods 1 and 2.

### Instrumentation

$^1$H-NMR spectra were obtained from solutions in DMSO-d$_6$, deuterium oxide, chloroform-d or methanol-d$_4$, and conducted on Bruker DPX-300, DPX-400, DPX-500 or DPX-600 spectrometers. The temperature was set at 20 °C. The chemical shift $\delta_H$ are given in parts per million using the signal for tetramethylsilane as reference. The coupling constants J are given in Hertz (Hz); the signals are described as follows: $s$ = singlet, $d$ = doublet, $t$ = triplet, $m$ = multiplet. $^{13}$C-NMR spectra were obtained from solutions in chloroform-d or methanol-d$_4$, and were conducted on Bruker DPX-300 or DPX-400 spectrometers. The temperature was set at 20 °C. The chemical shift $\delta_C$ are given in ppm relative to tetramethylsilane in DMSO-d$_6$. Mass spectra were recorded using a Waters GCT Premier (EI) or VG Quattro II TripleQuadrupole mass spectrometers (Electrospray). Mass spectrometry was performed

by Analytical Services and Environmental Projects (ASEP) at Queen's University, Belfast. Infrared spectra were recorded using a Perkin-Elmer Spectrum Two 91020 spectrophotometer in potassium bromide (KBr) disks. UV-Vis absorption spectroscopic measurements were made using an Agilent Cary 60 UV-Vis Spectrophotometer. Fluorescence emission spectroscopic measurements were made using a PerkinElmer LS55 Spectrometer. Surface tension measurements were made using an FTÅ200 instrument. The samples were loaded into a syringe and the syringe contents were pushed out by the instrument automatically. Photographs were taken when the drop was about to fall from the needle. All samples were tested ten times and an average was taken to improve the precision. Dynamic light-scattering measurements to measure the hydrodynamic diameters of the assemblies were conducted in aqueous solution at 25 °C with a Malvern Zetasizer Nano ZS. Nanoparticle tracking analysis measurements were made using a Malvern NanoSight NS300 instrument. The measurements were carried out at room temperature and laser type Blue 488 (nm) was used. Transmission electron microscopy was performed with a JEOL JEM-1400 transmission electron microscope operating at an accelerating voltage of 120 kV.

## Data availability

The authors declare that the data supporting the findings of this study are available within the article and its supplementary information files. All data are available from the corresponding author upon request.

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

## Acknowledgements

We thank the late Otto S. Wolfbeis and the late Sir J. Fraser Stoddart for support. We thank Queen's University Belfast, Department of Employment and Learning of Northern Ireland, Engineering and Physical Sciences Research Council UK, China Scholarship Council, the Leverhulme Trust (RPG-2019-314) and Ziwei Liu. E.V.A. thanks the Welch Regents Chair for support (F-0046).

## Author contributions

Z.Q.C., B.D., C.Y.Y., H.S.N.C., H.Q.N.G., A.K. and S.U. performed the experimental studies and contributed to data analysis. Y.X., Z.Y., S.E.J.B., E.V.A. and A.P.de S. supervised different aspects of the work. A.P.de S. conceived the project, analysed data and wrote the paper, with input from all co-authors.

## Competing interests

The authors declare no competing interests.
