## [Transparent Peer Review file · Nature Communications]

Scaling-up Molecular Logic to Meso-systems via Self-assembly

Corresponding Author: Professor A. Prasanna de Silva

Version 0:

Reviewer comments:

Reviewer #1

(Remarks to the Author)

The manuscript reports a series of cyclophanes as supramolecular switching elements in water. Logical operations are demonstrated on assembly of a multi-negatively charged cyclophane with a cationic detergent (or addition of a dicationic detergent). The assembly is indicated by turbidity in basic aqueous solution monitored by absorbance at 500 nm. TEM is used to rationalise the formation of multi-layer lamellae. The assemble is demonstrated as an integrated combinatorial logic array with a reset-set (RS) flip-flop memory component continuing a theme ref. 37-41, the first reported in this journal (Nature Commun. 10, 49, (2019)). A noteworthy result for supramolecular fans is that dimethylenecyclophane 1 and dialcoholcyclophane 3 accommodate the dication p-xylyldiammonium 9, while the more rigid diketonecyclophane 4 does not. Evidence is provided from NMR titration experiments in deuterated water. The study is a valued contribution to the fields of molecular logic-based computation and supramolecular chemistry. Acceptance is recommended after some minor revisions.

The studies were performed in basic aqueous solution (page 2, line 61) and the NMR captions indicate at pD 10 (Section S7). However, the introduction emphasises examples of computation in biological systems which typically occurs at pH levels about 7. How well do these cyclophane systems work at neutral physiological pH? Is hydroxide an additional (required) input for turbidity to be observed?

In the Introduction, it would be beneficial to readers to briefly define turbidity and what is meant by meso-scale.

It is claimed in the Introduction lines 37, 38 "that a membrane system is switchable in a Boolean manner for the first time" and that line 41 "membrane logic is unknown". A more thorough examination of the scientific literature likely provides examples in biological systems.

It should be reiterated more often in the manuscript that the study is perform in "water" or "aqueous solution" (i.e. page 3, line 123 when discussing binding constants).

Page 2, line 69: From the DLS data (Section S3) the maximum size peaks are 179 nm, 194 nm, 275 nm rather than 180 nm, 190 nm and 275 nm.

The manuscript lacks a section on the instrumentation and their specifications.

A photograph of the turbid solutions (1:9 ratio) versus control non-turbid solutions would be appealing to readers.

There are many ways of measuring turbidity including Environmental Protection standards. Please provide more details on how the data shown in Figure 1 was obtained.

The preparative procedures on page 2 of the SI cite ref. 35 and 38. It is preferable to again list these references in the supplementary information.

In the NMR characterisation data, the units of Hz should be added after the coupling constant values or at least a statement made in the Instrument section.

Reviewer #2

(Remarks to the Author)

The manuscript describes the successful realization of a meso-scaled self-assembled system, whose integral structural properties and assembly state are guided by redox inputs and supramolecular interaction with externally added cationic inputs. Indeed, as outlined by the authors, this fills a gap in the dimension scale of molecular logic gate systems. Their work builds on previously reported redox-induced switching of shape and geometry of anionic macrocycles, but is now combined with the up-scaling effect of their self-assembly into multilayered lamellae with cationic detergents, and the dissociative effect of external cationic inputs that depends on the specific structure of the assembly (deriving from the redox pair of macrocycles). This extends earlier reports by some of the authors in a very innovative manner and leads to complex logic functions that combine an R/S flip-flop memory with an extended combinatorial logic circuit. In my opinion this work is technically very well executed and the results support the drawn conclusions in full. The presented approach lifts molecular logic to an interesting spatial dimension. However, it would be nice if the authors could also include some comments on the potential use of their findings.

Reviewer #3

(Remarks to the Author)

In the article entitled "Scaling-up Molecular Logic to Meso-systems via Self-assembly", authors use self-assembly/collapse of anionic cyclophanes and cationic detergents to develop highly advanced membrane logic operation. Electrochemical pair of anionic cyclophanes (dialcohol and diketone) are elements of RS-flip flop and a combinational logic gate is constructed by xylyldiammonium cation. Turbidity is measured as ultimate output.

Although membrane embedded logic gates do exist, performing complex logic operations through manipulation of membrane forming abilities of small molecules is shown for the first time in this work, which would be an important milestone for scaling-up information processing self-assembled molecular systems. Following comments can be considered by the authors to clarify the manuscript:

1. Since the operation involves set of chemicals required for proper operation, the operational lifetime of the system might be limited. Is this logic device designed for single use? If not, accumulation of chemical waste and interference of the waste with the operation should be discussed.
2. Authors can discuss potential applications of such complex devices.
3. Critical micelle/aggregation concentrations of 10⁺(3 or 4) in the presence of compound 9 can be calculated to understand the interference of 9.
4. Authors should discuss the case when the redox reactions are not totally completed (indeed in SI page 4 yield for reoxidation of 3 is reported to be 85%).
5. Minor correction: Figure 3C, molecule name on the arrow should be 9.
6. Minor correction: In figure 4, names of the oxidizing and reducing agents should be given.

Version 1:

Reviewer comments:

Reviewer #1

(Remarks to the Author)

The corresponding author has taken the time to thoroughly address each of the Reviewer's queries. In particular, there is now enough detail in the methods for the work to be reproduced. A final suggestion is asked regarding Supplementary Fig. 2 where the molar fractions are given: to which of the two compounds do the number fractions apply?

Reviewer #3

(Remarks to the Author)

De Silva et. al. has taken into consideration and answered every question asked by the referees. A few experiments asked by the referees cannot be performed due to the retirement of the corresponding author and closure of his research lab. Considering that lack of these experiments does not reduce the significance of the research and considering that the research contributes a lot to information processing self-assembled molecular systems, I recommend the acceptance of article in this final form.

Reviewer #1 (Remarks to the Author)

The manuscript reports a series of cyclophanes as supramolecular switching elements in water. Logical operations are demonstrated on assembly of a multi-negatively charged cyclophane with a cationic detergent (or addition of a dicationic detergent). The assembly is indicated by turbidity in basic aqueous solution monitored by absorbance at 500 nm. TEM is used to rationalise the formation of multi-layer lamellae. The assembly is demonstrated as an integrated combinatorial logic array with a reset-set (RS) flip-flop memory component continuing a theme ref. 37-41, the first reported in this journal (Nature Commun. 10, 49, (2019)). A noteworthy result for supramolecular fans is that dimethylenecyclophane 1 and dialcoholcyclophane 3 accommodate the dication p-xylyldiammonium 9, while the more rigid diketonecyclophane 4 does not. Evidence is provided from NMR titration experiments in deuterated water. The study is a valued contribution to the fields of molecular logic-based computation and supramolecular chemistry. Acceptance is recommended after some minor revisions.

Response: We are grateful to Reviewer #1 for taking the trouble to understand what we tried to say. We appreciate his/her kind comments regarding about the value of our contribution.

The studies were performed in basic aqueous solution (page 2, line 61) and the NMR captions indicate at pD 10 (Section S7). However, the introduction emphasises examples of computation in biological systems which typically occurs at pH levels about 7. How well do these cyclophane systems work at neutral physiological pH? Is hydroxide an additional (required) input for turbidity to be observed?

Response: Reviewer #1 is correct that our work was conducted in basic aqueous solution. We did so in order to ensure that the cyclophanes 1, 3 & 4 remained in a near-maximally anionic state so that cationic guests like 10 and 9 would electrostatically attracted as strongly as possible. We note in passing that, being octabasic carboxylic acids, individual pKa values are difficult to extract for these cyclophanes [T. Swift, L. Swanson, M. Geoghegan & S. Rimmer, The pH-responsive behaviour of poly(acrylic acid) in aqueous solution is dependent on molar mass. *Soft Matter* 12, 2542–2549 (2016)], so that the degree of protonation at pH 7 would be hard to define. We did not attempt to study these systems at neutral pH, though we agree with Reviewer #1 that the correspondence of our work to biological systems is reduced as a result. Unfortunately, my recent retirement, the closure of the laboratory and the dispersal of samples and co-workers prevents us from conducting a parallel set of experiments at neutral pH. So we have added a sentence “However, the significance of our findings for biological systems needs to be tempered because these experiments were conducted in basic aqueous solution.” during revision on lines 232-234.

In the Introduction, it would be beneficial to readers to briefly define turbidity and what is meant by meso-scale.

Response: We appreciate Reviewer #1's suggestion to improve the benefit of our paper to readers. For the present purpose, the most convenient definition of turbidity is in terms of absorption of light (at a sufficiently long wavelength) caused by the scattering of light by the suspended particles. Since our compounds do not absorb light in the visible region, a wavelength of 500 nm was found to be practical. For the present purpose, the most convenient definition of meso-scale is the scale of sizes larger than nanometric and smaller than micrometric. These definitions have now been added on lines 67-69 and on lines 39-41.

It is claimed in the Introduction lines 37, 38 “that a membrane system is switchable in a Boolean manner for the first time” and that line 41 “membrane logic is unknown”. A more thorough examination of the scientific literature likely provides examples in biological systems.

Response: We are grateful to Reviewer #1 for encouraging us to hunt further. Indeed, there are examples of switchable membrane systems in biology. However, to the best of our knowledge, there have been no identifications of Boolean schemes corresponding to these

situations. So we have added the sentences “Conditional building-up and breakdown of membranes and proteins occurs continuously in nature, as seen during autophagy for instance. However, we are not aware of any Boolean schemes being identified in these situations.” during revision on lines 43-46. Indeed, we wish that our work would encourage biological scientists to think of membrane phenomena in terms of computing operations.

The reference list has been expanded with;

36. Y. Ohsumi, The Nobel Prize in Physiology and Medicine, 2016. The Nobel Prize, <https://www.nobelprize.org/prizes/medicine/2016/ohsumi/lecture/>

Also, we are not aware of any synthetic membrane switchable in a Boolean manner except for a single case published after this manuscript was submitted. The logic arrays in this case are simpler than what is described in the present work. Accordingly, we have rephrased the sentences in the Introduction lines 37, 38 (numbering in original manuscript now appears in the revised manuscript on lines 37,38) as “We now show that a membrane system is switchable in a Boolean manner involving more than 10 logic elements” and lines 41,42 (now appears in the revised manuscript on line 43) as “membrane logic is very rare”.

The reference list has been expanded with;

35. J. H. Wei, J. F. Xing, X. F. Hou, X. M. Chen & Q. Li, Light-operated diverse logic gates enabled by modulating time-dependent fluorescence of dissipative self-assemblies. *Adv. Mater.* 2411291 (2024).

We have also added the sentences “It is also relevant to note a report⁶⁵ where the concept of Boolean drug release following depolymerization of materials^{33,34} has been miniaturized to the nanoparticle level. However, the nanoparticles are irreversibly degraded during the release process.” in the revised manuscript on lines 163-166.

The reference list has been expanded with;

65. P. H. Zhang, D. Gao, K. L. An, Q. Shen, C. Wang, Y. C. Zhang, X. S. Pan, X. G. Chen, Y. F. Lyv, C. Cui, T. X. Z. Liang, X. M. Duan, J. Liu, T. L. Yang, X. X. Hu, J. J. Zhu, F. Xu & W. H. Tan, A programmable polymer library that enables the construction of stimuli-responsive nanocarriers containing logic gates. *Nat. Chem.* 12, 381-390 (2020).

It should be reiterated more often in the manuscript that the study is performed in “water” or “aqueous solution” (i.e. page 3, line 123 when discussing binding constants).

Response: We are grateful to Reviewer #1 for this advice which will surely increase the relevance of our paper to readers. We have inserted the phrases ‘in water’ or ‘in aqueous solution’ at lines 85, 90, 134, 138, 173, 246, 468, 473 and 491. Similar adjustments have been made to captions to Supplementary Figs. 4a, 7 and the header in Supplementary Table 1 in the Supplementary Information.

Page 2, line 69: From the DLS data (Section S3) the maximum size peaks are 179 nm, 194 nm, 275 nm rather than 180 nm, 190 nm and 275 nm.

Response: Although we previously gave rounded values in-line with estimated errors, we bow to Reviewer #1’s suggestion and give the values reported by the DLS instrument.

The manuscript lacks a section on the instrumentation and their specifications.

Response: We are happy to follow Reviewer #1’s advice. We now give a section on the instruments employed and their specifications in Supplementary methods 2.

A photograph of the turbid solutions (1:9 ratio) versus control non-turbid solutions would be appealing to readers.

Response: We appreciate this suggestion by Reviewer #1. The only photographs available to the corresponding author (who is now retired) are those of a preliminary experiment

involving **1** and **10**. These now appear in the revised manuscript as Supplementary Fig. 2. Hopefully, these will give the readers a feel for what was observed, although their conditions are not exactly those encountered in Figure 1c.

There are many ways of measuring turbidity including Environmental Protection standards. Please provide more details on how the data shown in Figure 1 was obtained.

Response: Reviewer #1 is indeed correct that turbidity is used in a specific quantitative way within environmental science. For example, nephelometers measure low levels of turbidity and absorptimeters measure high levels in terms of absolute values for environmental standards. We now realize that our approach to turbidity from a chemistry viewpoint would be confusing to environmental scientists, for instance, unless it is clearly defined. We thank Reviewer #1 for this insight. So we have defined turbidity in our experiments on lines 67-70 as the absorbance in a cell of 1 cm optical path measured at a wavelength which none of the components absorb, which was chosen as 500 nm. This is sufficient for our work since we only needed a semi-quantitative measure to show when the turbidity maximized.

The preparative procedures on page 2 of the SI cite ref. 35 and 38. It is preferable to again list these references in the supplementary information.

Response: We thank Reviewer #1 for this suggestion which improves the clarity of our paper and its supporting information. References 35 and 38 have been given again in the supporting information within the Supplementary reference list.

In the NMR characterisation data, the units of Hz should be added after the coupling constant values or at least a statement made in the Instrument section.

Response: We thank Reviewer #1 for this suggestion, which has been done after each coupling constant value.

Reviewer #2 (Remarks to the Author)

The manuscript describes the successful realization of a meso-scaled self-assembled system, whose integral structural properties and assembly state are guided by redox inputs and supramolecular interaction with externally added cationic inputs. Indeed, as outlined by the authors, this fills a gap in the dimension scale of molecular logic gate systems. Their work builds on previously reported redox-induced switching of shape and geometry of anionic macrocycles, but is now combined with the up-scaling effect of their self-assembly into multilayered lamellae with cationic detergents, and the dissociative effect of external cationic inputs that depends on the specific structure of the assembly (deriving from the redox pair of macrocycles). This extends earlier reports by some of the authors in a very innovative manner and leads to complex logic functions that combine an R/S flip-flop memory with an extended combinatorial logic circuit. In my opinion this work is technically very well executed and the results support the drawn conclusions in full. The presented approach lifts molecular logic to an interesting spatial dimension.

Response: We thank Reviewer #2 for not only appreciating what we tried to say in our paper, but also for seeing it in the context of our older efforts. We appreciate his/her kind comments regarding our paper.

However, it would be nice if the authors could also include some comments on the potential use of their findings.

Response: We thank Reviewer #2 for giving us this opportunity. Reviewer #3 raised this point as well. A potential use of our findings on membrane logic systems would be in the diagnosis of membrane-related diseases [C. Dias & J. Nylandsted, Plasma membrane integrity in health and disease: Significance and therapeutic potential. *Cell Discovery* 7, 4 (2021)] just as nanometric logic systems were proposed for diseases concerning electrolytes (A. J. Bryan, A. P. de Silva, S. A. de Silva, R. A. D. D. Rupasinghe & K. R. A. S. Sandanayake, Photo-induced electron transfer as a general design logic for fluorescent

molecular sensors for cations. *Biosensors* 4, 169-179 (1989); D. C. Magri, G. J. Brown, G. D. McClean & A. P. de Silva, Communicating chemical congregation: A molecular and logic gate with three chemical inputs as a “lab-on-a-molecule” prototype. *J. Am. Chem. Soc.* 126, 4950-4951 (2006)) and commercialized later for blood diagnostics (www.optimedical.com). This has been added at lines 230-242. These four references have been added to the reference list at numbers 72-75.

Reviewer #3 (Remarks to the Author):

In the article entitled “Scaling-up Molecular Logic to Meso-systems via Self-assembly”, authors use self-assembly/collapse of anionic cyclophanes and cationic detergents to develop highly advanced membrane logic operation. Electrochemical pair of anionic cyclophanes (dialcohol and diketone) are elements of RS-flip flop and a combinational logic gate is constructed by xylyldiammonium cation. Turbidity is measured as ultimate output. Although membrane embedded logic gates do exist, performing complex logic operations through manipulation of membrane forming abilities of small molecules is shown for the first time in this work, which would be an important milestone for scaling-up information processing self-assembled molecular systems.

Response: We thank Reviewer #3 for appreciating how our paper advances information processing self-assembled molecular systems.

Following comments can be considered by the authors to clarify the manuscript:

1. Since the operation involves set of chemicals required for proper operation, the operational lifetime of the system might be limited. Is this logic device designed for single use? If not, accumulation of chemical waste and interference of the waste with the operation should be discussed.

Response: We agree with Reviewer #3 that chemically-switchable logic systems do accumulate chemical waste. However, there are examples where waste does not interfere with their operation at least in the short term (ref. 40). Ref. 40 is also an example closely related to the present work where three cycles of operation have been run successfully. However, the present work is currently validated for single use. This has been added at lines 236-238.

2. Authors can discuss potential applications of such complex devices.

Response: Reviewer #2 mentioned this too. A potential application of membrane logic systems would be in the diagnosis of membrane-related diseases [C. Dias & J. Nylandsted, Plasma membrane integrity in health and disease: Significance and therapeutic potential. *Cell Discovery* 7, 4 (2021)] just as nanometric logic systems have been proposed for diseases concerning electrolytes [A. J. Bryan, A. P. de Silva, S. A. de Silva, R. A. D. D. Rupasinghe & K. R. A. S. Sandanayake, Photo-induced electron transfer as a general design logic for fluorescent molecular sensors for cations. *Biosensors* 4, 169-179 (1989); D. C. Magri, G. J. Brown, G. D. McClean & A. P. de Silva, Communicating chemical congregation: A molecular and logic gate with three chemical inputs as a “lab-on-a-molecule” prototype. *J. Am. Chem. Soc.* 126, 4950-4951 (2006)] and commercialized for blood diagnostics (www.optimedical.com). This has been added at lines 230-232. These four references have been added to the reference list at numbers 72-75.

3. Critical micelle/aggregation concentrations of 10+(3 or 4) in the presence of compound 9 can be calculated to understand the interference of 9.

Response: We agree that Reviewer #3's suggestion of measuring critical aggregation concentrations of the surfactant **10 + (3 or 4)** in the presence of compound **9** would be very worthwhile, especially because we had reported the critical aggregation concentrations of the surfactant **10 + (3 or 4)** in Supplementary Table 1. Unfortunately, the closure of the laboratory prevents us from performing this experiment. While apologizing for this turn of

events, we could add a sentence 'It would have been informative to measure the critical aggregation concentrations of the surfactant **10** + (**3** or **4**) in the presence of compound **9**, but circumstances have prevented us from doing so' on line 156. However, such a sentence might be scientifically meaningless to a reader who is unaware of the laboratory closure. If Reviewer #3 and the Editor are agreeable, we would prefer not to add this sentence.

4. Authors should discuss the case when the redox reactions are not totally completed (indeed in SI page 4 yield for reoxidation of **3** is reported to be 85%).

Response: Reviewer #3 is perceptive in noticing that the preparative redox reactions in the present work are not quantitative. A similar situation arose in ref. 40, where the large-scale preparative procedures that gave 88% and 80%. However, we were able to optimize them at a 50-100 mg scale to give yields of 94% and 95%. We feel that a similar optimization of reaction conditions should be possible here too. Since we are unable to carry this through owing to the closure of the laboratory, it likely that the sharpness of switching of turbidity (Figure 4A) will be tempered somewhat owing to the current situation of incomplete redox reactivity. The sentences 'A further limitation is introduced by the fact that the redox interconversions of **3** and **4** are not quantitative at present. The sharpness of switching of turbidity (Figure 4A) will be tempered somewhat owing to this.' have been added at line 240.

5. Minor correction: Figure 3C, molecule name on the arrow should be **9**.

Response: We are grateful to Reviewer #3 for catching this error. This has now been corrected.

6. Minor correction: In figure 4, names of the oxidizing and reducing agents should be given.

Response: We have happily followed Reviewer #3's advice on this point.

Reviewer #1 (Remarks to the Author):

The corresponding author has taken the time to thoroughly address each of the Reviewer's queries. In particular, there is now enough detail in the methods for the work to be reproduced. **Response: We are pleased that Reviewer #1 is satisfied with the revisions.**

A final suggestion is asked regarding Supplementary Fig. 2 where the molar fractions are given: to which of the two compounds do the number fractions apply?

Response: We thank Reviewer #1 for pointing out this final revision for clarity. The caption of Supplementary Fig. 2 has been adjusted to 'Photographs of aqueous solutions formed by mixtures of **1 and **10** with the following molar fractions of **10** (X_{10})'.**

Reviewer #3 (Remarks to the Author):

De Silva et. al. has taken into consideration and answered every question asked by the referees. A few experiments asked by the referees cannot be performed due to the retirement of the corresponding author and closure of his research lab. Considering that lack of these experiments does not reduce the significance of the research and considering that the research contributes a lot to information processing self-assembled molecular systems, I recommend the acceptance of article in this final form.

Response: We are pleased that Reviewer #3 is satisfied with almost all the revisions. We are grateful to Reviewer #3 for his/her understanding of our situation regarding the retirement of the corresponding author which prevented us from carrying out the revision concerning the requested critical micelle concentration determination.